# ASMT determines gut microbiota and increases neurobehavioral adaptability to exercise in female mice

Weina Liu [1,2,3✉], Zhuochun Huang[1,2], Ye Zhang[1,2], Sen Zhang[1,2], Zhiming Cui[1,2], Wenbin Liu[1,2], Lingxia Li[1,2], Jie Xia[1,2], Yong Zou[1,2] & Zhengtang Qi [1,2,3✉]

N-acetylserotonin O-methyltransferase (ASMT) is responsible for melatonin biosynthesis. The *Asmt* gene is located on the X chromosome, and its genetic polymorphism is associated with depression in humans. However, the underlying mechanism remains unclear. Here, we use CRISPR/Cas9 to delete 20 bp of exon 2 of *Asmt*, and construct *C57BL/6J* mouse strain with *Asmt* frameshift mutation (*Asmt*^ft/ft). We show that female *Asmt*^ft/ft mice exhibit anxiety- and depression-like behaviors, accompanied by an obvious structural remodeling of gut microbiota. These behavioral abnormalities are not observed in male. Moreover, female *Asmt*^ft/ft mice show a lower neurobehavioral adaptability to exercise, while wild-type shows a "higher resilience". Cross-sectional and longitudinal analysis indicates that the structure of gut microbiota in *Asmt*^ft/ft mice is less affected by exercise. These results suggests that *Asmt* maintains the plasticity of gut microbiota in female, thereby enhancing the neurobehavioral adaptability to exercise.

[1] The Key Laboratory of Adolescent Health Assessment and Exercise Intervention (Ministry of Education), East China Normal University, Shanghai 200241, China. [2] College of Physical Education and Health, East China Normal University, Shanghai 200241, China. [3] These authors contributed equally: Weina Liu, Zhengtang Qi. ✉email: wnliu@tyxx.ecnu.edu.cn; ztqi@tyxx.ecnu.edu.cn

About 27.2% of women and 7–12% of men suffer from depression according to World Health Report issued by WHO, suggesting that women are more prone to depression than men[1]. Social and psychological stress has always been identified as the root causes of anxiety and depression in humans, so animal studies have established rodent models with social defeat[2] and chronic stress[3]. Male animals are thought to avoid interference from the estrous cycle, so females have to be left out of most previous studies. Studies favoring males overlooked an important question: Why are females prone to depression? Are there inherent genetic traits that make females more susceptible to depression or mood disorders? Differential expression of sex-related genes throughout life may be implicated in depression vulnerability. There are undeniable differences in how men and women cope with social stress and frustration. On the other hand, the effects of treatments or medications that improve depression and anxiety are also gender-related[4]. Therefore, studies on the interaction between gender and environment are needed to unravel the complexity of the causes of depression[5].

Poor sleep quality may contribute to depressive symptoms[6], thus stimulating neurogenesis and improving sleep with melatonin is beneficial for depressed patients[7] and mice[8]. The *C57BL/6J* inbred mouse strain is the most preferred choice for behavioral studies[9]. Although the *C57BL/6J* mouse strain is inborn with melatonin deficiency in the pineal gland[10], melatonin is also produced in other tissues (such as the cerebral cortex, serum, heart, liver, and kidney) besides the pineal gland[11,12]. Arylalkylamine N-acetyltransferase (AANAT) and N-acetylserotonin O-methyltransferase (ASMT) enzyme activity affects the concentration of N-acetylserotonin, which determines the circulating level of melatonin[13]. Mouse *Asmt* gene (ID:107626) is located in the F5 region of X chromosome (NC_000086.8)[14], moreover, psychiatry-related behaviors are relatively consistent throughout the estrous cycle in female *C57BL/6J* mice[15]. Zhang et al. generated a congenic line of the *C57BL/6J* mouse strain with the capacity of melatonin synthesis via introducing functional alleles of *Aanat* and *Asmt* genes from the melatonin-proficient CBA/CaJ mouse strain to B6[16], and genetic deletion of MT1 melatonin receptor caused depressive and anxiety-like behaviors in male and female C57BL/6 mice[17].

Here, we found that *Asmt* frameshift mutation (*Asmt*^ft/ft) caused anxiety- and depression-like behaviors in female mice, but not in males. Physical inactivity has been found to amplify the negative association between depression and sleep quality in adult participants[18], and endurance exercise can improve depression-like behaviors in rats with pinealectomy[19]. Voluntary exercise has been reported to elevate key proteins involved in synaptic plasticity and improve gut microbiota profiles, and thus prevent cognitive decline in mice[20]. In this study, our findings showed that *Asmt*^ft/ft enhanced hippocampal neuroplasticity for exercise. As a consequence, *Asmt*^ft/ft impaired the adaptability of gut microbiota and thus reduced the neurobehavioral adaptations to exercise in the *C57BL/6J* mice.

## Results

**Female *Asmt*^ft/ft mice exhibits anxiety- and depression-like behaviors**. We first examined the anxiety- and depression-like behaviors in mice. Our results showed that female *Asmt*^ft/ft mice had less distance in center ($p < 0.001$) and a smaller number of poking ($p < 0.05$) in the open field test (OFT, Fig. 1a–c). In the forced swim test (FST), immobility time was higher ($p < 0.05$) whereas struggling time was lower ($p < 0.05$) in female *Asmt*^ft/ft mice compared with wt/wt mice (Fig. 1d, e). Behavioral tests further indicated that female *Asmt*^ft/ft mice spent more immobility time in tail suspension test (TST) than that of wt/wt mice

($p < 0.05$, Fig. 1f, g). On the contrary, male *Asmt*^ft/ft mice had more distance in center ($p < 0.05$) and a greater number of poking ($p < 0.05$) in OFT (Fig. 1a, b), and spent more struggling time in the TST than that of wt/wt mice ($p < 0.05$, Fig. 1g). However, sucrose preference test (SPT) did not show behavioral abnormalities (Fig. 1h).

All behavioral data were further analyzed by nested *t*-test. Results of mixed sex comparison showed no behavioral differences between wt/wt and ft/ft mice, with *P*-values between 0.3489 and 0.9638, suggesting that these differences were not statistically significant (Fig. S1a–h). However, Chi-square test results showed that there was a greater gender difference in the OFT and FST (Fig. S1a–e). Therefore, when we explore the genotype effects of *Asmt*, gender must be considered. That is, *Asmt*^ft/ft increased the anxiety- and depression-like behaviors only in female *C57BL/6J* mice. These behavioral results led us to include only female mice in the following studies.

**Asmt^ft/ft attenuates neurobehavioral adaptability to exercise**. Female mice were grouped and subjected to exercise training for a total duration of 8 weeks before behavioral tests (Fig. 1i). Our results suggest that *Asmt*^ft/ft genotype had different neurobehavioral adaptability to exercise compared to wt/wt mice. In OFT (Fig. 2a–c), there was a significant (genotype × exercise) interaction for distance in the center ($p < 0.01$), whereas exercise increased poking number in ft/ft mice ($p < 0.05$), and reduced time in center in wt/wt mice ($p < 0.05$). In FST (Fig. 2d, e), there was a significant (genotype × exercise) interaction for struggling time ($p < 0.05$). The ft/ft mice did not show significant changes in immobility time or struggling time after exercise, whereas exercise increased immobility time ($p < 0.05$) and decreased struggling time ($p < 0.05$) in wt/wt mice. In TST (Fig. 2f, g), we also found a significant (genotype × exercise) interaction for struggling time ($p < 0.01$). Exercise increased immobility time ($p < 0.01$) and decreased struggling time ($p < 0.05$) in wt/wt mice, whereas ft/ft mice behaviors showed no response to exercise in TST. In addition, wt/wt mice displayed a greater increase in immobility time ($p < 0.01$, 101%) compared to ft/ft mice (5.5%) following exercise (Fig. 2f). No significant differences were observed in the SPT (Fig. 2h). These results suggest that the behavioral effects of exercise are strongly determined by genotype in female mice.

**Asmt^ft/ft enhances hippocampal neuroplasticity for exercise**. Synaptic plasticity in hippocampus was a neurobiological mechanism linked to brain memory and depression[21,22], so we next evaluated the immunoreactivity of plasticity-related protein in CA1 and DG hippocampal regions. In the CA1v, immunofluorescent images showed that exercise increased GAP-43 expression ($p < 0.05$, Fig. 3a) and astrocytes number labeled by GFAP ($p < 0.05$, Fig. 3c) in the ft/ft group. Moreover, exercise also increased the NeuN immunoreactivity in the CA1d region of ft/ft mice ($p < 0.05$, Fig. 3b). On the contrary, no significant difference was observed in CA1 between the wt/wt and Ex^wt/wt group (Fig. 3a–c). In the DG, ft/ft mice showed a lower expression of GAP-43 ($p < 0.01$, Fig. 3d) and an increased number of astrocytes labeled by GFAP ($p < 0.05$, Fig. 3f) compared to wt/wt mice. On the contrary, Ex^ft/ft mice showed a higher expression of GAP-43 ($p < 0.05$, Fig. 3d) and SYP ($p < 0.05$, Fig. 3e) but decreased number of astrocytes labeled by GFAP ($p < 0.05$, Fig. 3f) compared to ft/ft mice. Similarly, no significant difference was observed in DG between the wt/wt and Ex^wt/wt groups (Fig. 3d–f).

Electronic microscope images and quantitative analysis showed that *Asmt*^ft/ft increased the number of synapses in hippocampal CA1 region ($p < 0.05$), but exercise had no significant effect on synapses number in wt/wt or ft/ft mice (Fig. 3g). In addition,

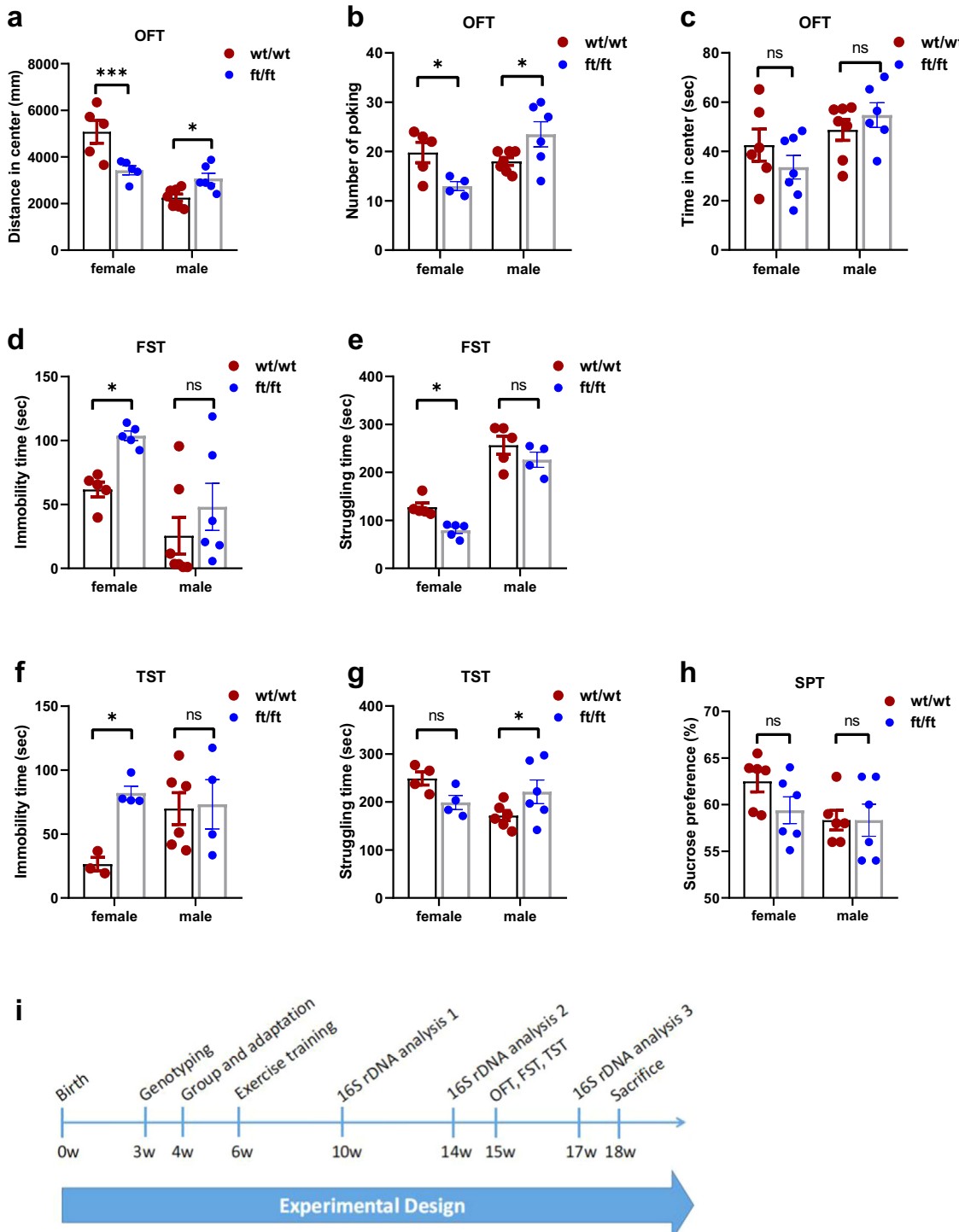

**Fig. 1 *Asmt* frameshift mutation (*Asmt*^ft/ft^) induced anxiety and depression-like behaviors in female *C57BL/6J* mice. a–c** Distance in center, number of poking, and time in center in the OFT ($n = 4$–7/group; **a** Two-way ANOVA, $F_{(1,19)} = 20.27$, $p = 0.0002$. **b** Two-way ANOVA, $F_{(1,18)} = 11.19$, $p = 0.0036$. **c** Two-way ANOVA, $F_{(1,22)} = 2.156$, $p = 0.1562$). **d, e** Immobility time and struggling time in the FST ($n = 4$–7/group). **d** Two-way ANOVA, $F_{(1, 19)} = 11.45$, $p = 0.0031$. **e** Two-way ANOVA, $F_{(1,15)} = 107.0$, $p < 0.0001$. **f, g** Immobility time and struggling time in the TST ($n = 4$–7/group; **f** Two-way ANOVA, $F_{(1,13)} = 4.712$, $p = 0.0491$. **g** Two-way ANOVA, $F_{(1,16)} = 7.237$, $p = 0.0161$. **h** Sucrose preference in the SPT ($n = 6$/group; Two-way ANOVA, $F_{(1,20)} = 1.289$, $p = 0.2696$). **i** Experimental schedule (18 weeks). Briefly, mice were genotyped at 3 weeks of age and grouped according to sex and genotype at 4 weeks of age. At 6 weeks of age, the mice were already well-equipped to exercise. Feces were collected for gut microbiota analysis during the 4th and 8th week of exercise and in the third week after exercise (i.e., after behavioral tests). Behavioral tests were performed at 15–17 weeks of age, and sacrifices were made at 18 weeks of age. Data are mean ± SEM. *$p < 0.05$, ***$p < 0.001$; ns, no significance. Two-way ANOVA followed by post hoc test.

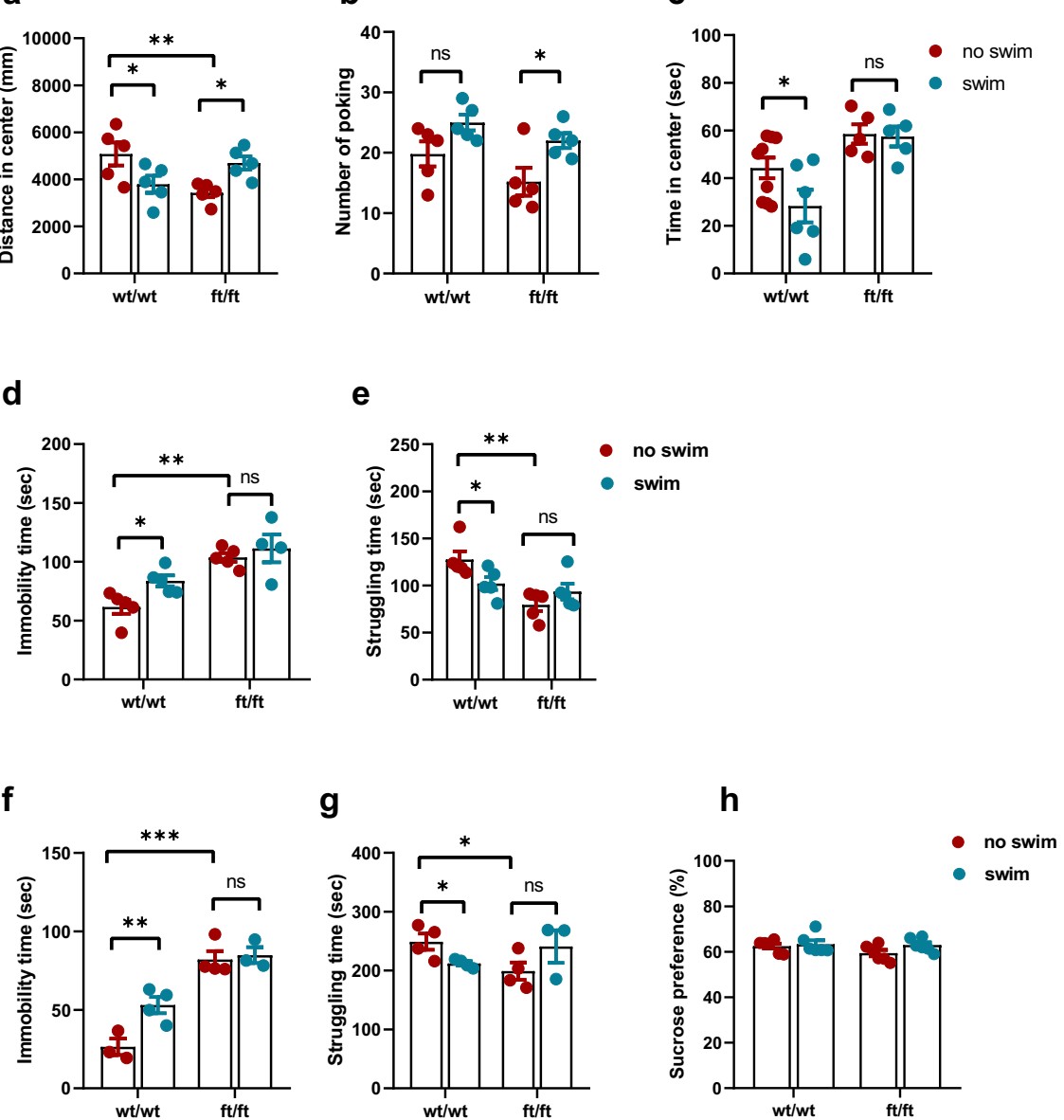

**Fig. 2 *Asmt*<sup>ft/ft</sup> altered the behavioral adaptation of mice to exercise. a–c** Distance in the center, number of poking, and time in center in the OFT ($n = 5$–7/group; **a** Two-way ANOVA, $F_{(1,16)} = 13.26$, $p = 0.0022$; **b** Two-way ANOVA, $F_{(1,16)} = 4.484$, $p = 0.0502$; **c** Two-way ANOVA, $F_{(1,21)} = 16.89$, $p = 0.0005$). **d, e** Immobility time and struggling time in the FST ($n = 4$–6/group; **d** Two-way ANOVA, $F_{(1,16)} = 11.32$, $p = 0.0039$; **e** Two-way ANOVA, $F_{(1,16)} = 6.686$, $p = 0.0199$). **f, g** Immobility time and struggling time in the TST ($n = 3$–5/group; **f** Two-way ANOVA, $F_{(1, 10)} = 4.952$, $p = 0.0502$; **g** Two-way ANOVA, $F_{(1,11)} = 6.533$, $p = 0.0267$). **h** Sucrose preference in the SPT ($n = 6$/group; Two-way ANOVA, $F_{(1,20)} = 0.9922$, $p = 0.3311$). Data are mean ± SEM. \**p* < 0.05, \*\**p* < 0.01, \*\*\**p* < 0.001; ns, no significance. Two-way ANOVA followed by post hoc test.

*Asmt*<sup>ft/ft</sup> genetic manipulation reduced PSD thickness in hippocampal CA1 region ($p < 0.05$) and exercise reversed this change in synaptic structure ($p < 0.01$), whereas exercise had no significant effect on PSD thickness in wt/wt mice (Fig. 3h). Together, *Asmt*<sup>ft/ft</sup> impairs neurogenesis and synaptic structure in the hippocampus, and exercise can reverse these changes.

**Asmt<sup>ft/ft</sup> reduces gut microbiotal adaptations to exercise: a cross-sectional comparison at the endpoint.** We analyzed the fecal microbiota profiles of all groups by bacterial taxa *16S* rRNA amplicon sequencing from feces collected at 3 weeks post-exercise (after the behavioral test). Principal component analysis (PCoA) revealed a trend of separation between ft/ft and wt/wt genotypes,

and running exercise induced a smaller trend of separation in ft/ft mice than that in wt/wt mice (Fig. 4a).

At the species level, ft/ft mice showed a higher abundance of *Faecalibaculum_rodentium* ($p < 0.05$), *Dorea_sp._5-2* ($p < 0.05$), *Firmicutes_bacterium_CAG_194_44_15* ($p < 0.05$) and *Lactobacillus_salivarius* ($p < 0.05$), and a lower abundance of *Bacteroides_fragilis* ($p < 0.01$), *Firmicutes_bacterium_M10-2* ($p < 0.01$), *Desulfovibrionales_bacterium* ($p < 0.01$), Mucispirillum_schaedleri_ASF457 ($p < 0.05$), *Lactobacillus_murinus* ($p < 0.05$) compared to wt/wt mice (Figs. 5a and S2a). In the *Asmt*<sup>ft/ft</sup> mice, exercise increased the abundance of *human_gut_metagenome* ($p < 0.01$), *Lactobacillus_mucosae* ($p < 0.01$), *Firmicutes_bacterium_M10-2* ($p < 0.01$), *Burkholderiales_bacterium_YL45* ($p < 0.05$), *Bacteroides_caecimuris* ($p < 0.05$), *Mucispirillum_schaedleri_ASF457*

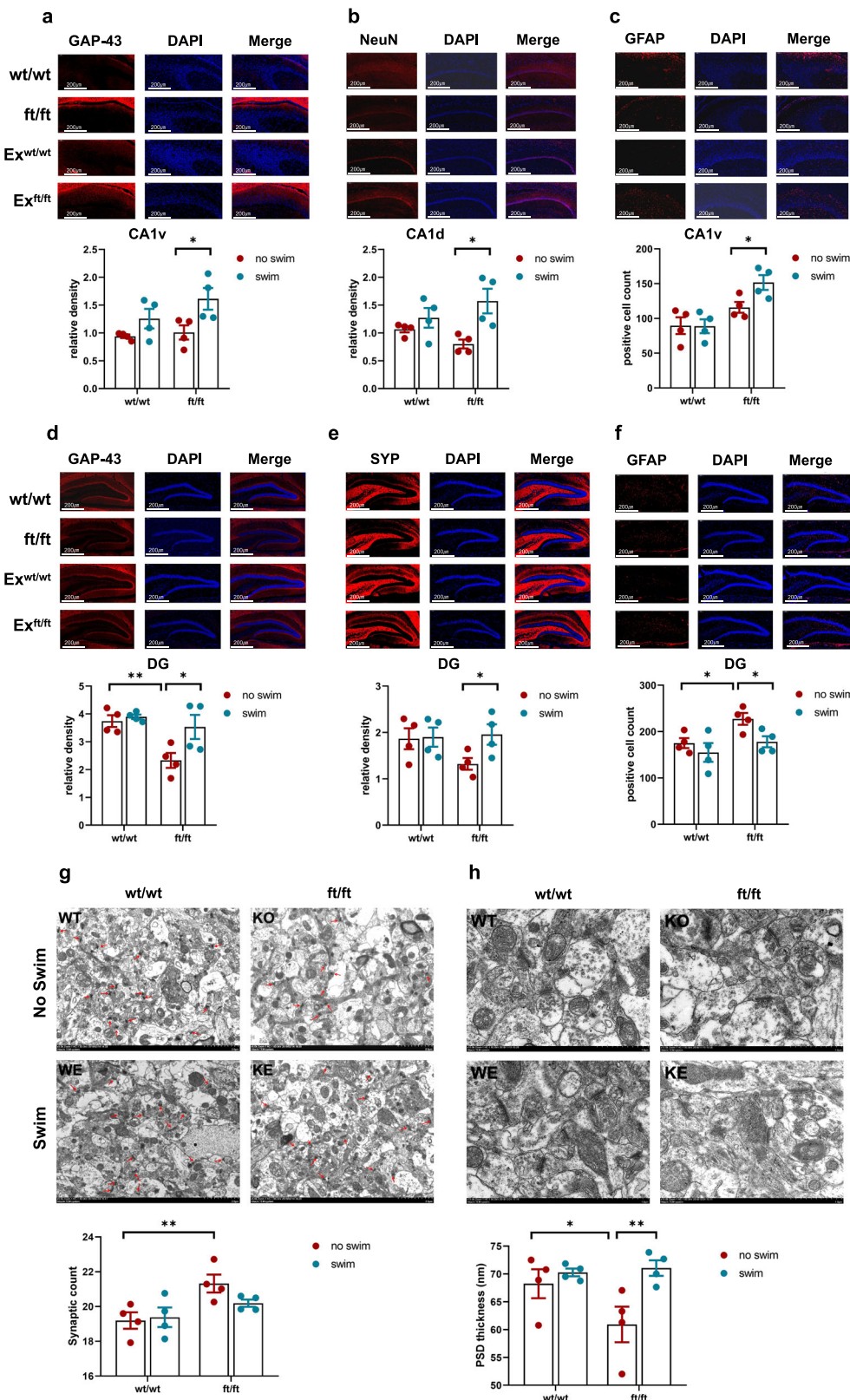

(p < 0.05), *Desulfovibrionales_bacterium* (p < 0.05), and reduced the abundance of *Bacteroides_fragilis* (p < 0.05), *Lactobacillus_-fermentum* (p < 0.05) compare to ft/ft mice (Figs. 5a and S2b). In the wt/wt mice, exercise increased the abundance of *Bacteroidales_bacterium* (p < 0.05), *Burkholderiales_bacterium_YL45* (p < 0.01), *Gemmatimonas_sp.* (p < 0.05), *Lactococcus_ lactis* (p < 0.05), *Spiroplasma_leptinotarsae* (p < 0.05), and *reduced the abundance of Bacteroides_fragilis* (p < 0.05), *Clostridiales_bacterium* (p < 0.05), *Lactobacillus_murinus* (p < 0.05) compare to wt/wt mice (Figs. 5a and S2c). At the phylum, family, and genus level, we also found that there was interaction between genotype and exercise on the remodeling of gut microbiota. Figure 5b–g

**Fig. 3 $Asmt^{ft/ft}$ mice displayed higher synaptic plasticity during exercise than wild-type. a–c** GAP-43, NeuN, and the astrocytes number labeled by GFAP in hippocampal CA1 region. Immunofluorescence images are shown above and quantification is shown below ($n = 4$/group; **a** Two-way ANOVA, $F(1, 12) = 0.9531$, $p = 0.3482$; **b** Two-way ANOVA, $F(1, 12) = 3.559$, $p = 0.0837$; **c** Two-way ANOVA, $F(1, 12) = 3.217$, $p = 0.0981$. **d–f** GAP-43, SYP, and the astrocytes number labeled by GFAP in hippocampal DG region ($n = 4$/group; **d** Two-way ANOVA, $F(1, 12) = 3.523$, $p = 0.0850$; **e** Two-way ANOVA, $F(1, 12) = 2.280$, $p = 0.1570$; **f** Two-way ANOVA, $F(1, 12) = 1.039$, $p = 0.3282$). **g, h** The number of synapses and thickness of PSD in Electron microscopy ($n = 4$/group; **g** Two-way ANOVA, $F(1, 12) = 2.051$, $p = 0.1776$; **h** Two-way ANOVA, $F(1, 12) = 3.406$, $p = 0.0897$). Data are mean ± SEM. Genotype effect, $*p < 0.05$, $**p < 0.01$; Exercise effect: $*p < 0.05$, $**p < 0.01$; Two-way ANOVA followed by post hoc test. CA1d dorsal hippocampus, CA1v ventral hippocampus, DG hippocampal dentate gyrus, PSD postsynaptic density.

showed that exercise only had a significant effect on the relative abundance of gut microbiota in wild-type mice, not in ft/ft mice, at any level.

In the top 30 different species, the operational taxonomic units (OTUs) of *Mucispirillum* (OTU172), as well as *Lachnospiraceae* (OTU13), were negatively correlated with 14 OTUs and positively correlated with 10 OTUs (Fig. 4b). *Bacteroides* (OTU84) was negatively correlated with 12 OTUs and positively correlated with 8 OTUs (Fig. 4b). To determine the correlation between behavioral parameters and changes in species level, we performed permutation analysis of variance. Two indicators of depressive behavior were correlated with the abundance of *Alistipes* and *Mucispirillum* in the gut microbiota: immobility time and struggling time in FST (Fig. 4c).

Linear discriminant analysis effect size (LEfSe) was employed to identify specific species responding to exercise in both wt/wt and ft/ft genotypes. In wt/wt mice, 59 species at the OTU level were identified as high-dimensional biomarkers for separating gut microbiota between sedentary and running groups (Fig. S3). 39 of these OTUs were higher, and 20 were lower in the exercise group than in the sedentary group. For example, the abundances of *Firmicutes* and *Lachnospiraceae* were lower after exercise. In ft/ft mice, 48 OTUs were higher and 16 were lower in the running group than in the sedentary group (Fig. S4). More than 90% of these OTUs were not identical to those identified in wt/wt mice, indicating that gut microbiotic reconstitution for exercise in ft/ft mice was different from that in wt/wt mice. The mice had a difference in gut microbiota structure between the wt/wt and ft/ft groups, confirming the PCA results in Fig. 4a. 85 species were selected as key variables for separating the gut microbiota by genotypes (Fig. S5), showing that exercise reconstituted the gut microbiota profiles in wt/wt mice, not in the ft/ft mice.

### $Asmt^{ft/ft}$ suppresses gut microbial remodeling during exercise: a longitudinal analysis.

*16S rRNA* amplicon sequencing was performed on feces collected at the 4th and 8th weeks of exercise and at the end of the behavioral test (~3 weeks post-exercise). Running exercise obviously influenced the richness and diversity of the bacterial community in wt/wt mice, but not in ft/ft mice. Disturbances associated with running exercise were indicated by plots of Shannon index, Simpson index, OTU number, and PD whole tree over time. Compared with the sedentary group, these four indexes showed no significant change during the first four weeks of exercise in either wt/wt or ft/ft group (Fig. 6a–d). This suggests that 4-week exercise had no significant effect on the structure of gut microbes. However, the differences between wt/wt and ft/ft groups expanded after the 4th week. OTU richness and diversity increased gradually in the wt/wt group and reached the highest value in the 3rd week after exercise (OTU number, $p < 0.05$; Shannon and Simpson index, $p < 0.01$). These parameters in ft/ft mice showed no significant changes at the 8th week of exercise and the 3rd week after exercise (Fig. 6a–d). A greater difference was found in the PD whole tree, which remained stable at the 8th week of exercise but decreased significantly at the 3rd week after exercise in ft/ft mice ($p < 0.01$, red asterisk), while

increased significantly at the 8th week of exercise in wt/wt mice ($p < 0.05$, black asterisk) and remained stable at the 3rd week after exercise (Fig. 6d). These results indicated that the richness and diversity of gut microbiota did not increase or decrease linearly during exercise, and its structure was greatly influenced by genotype.

Time-course changes in gut microbial structures of mice during exercise were assessed by PCoA of weighted UniFrac distances. The structure of the gut microbiota at each time point exhibited no significant differences among ft/ft animals. However, the PCoA score plot showed a separation of gut microbiota structure in wt/wt mice (Fig. 6g, h). Among the top 15 different phyla, exercise had a greater effect on wt/wt mice than ft/ft mice. Specifically, the three phyla with the highest relative abundance in the two genotypes were identical, namely *Bacteroidetes, Firmicutes,* and *Proteobacteria*. The colors of the three phyla clearly showed that the gut microbiota structure of ft/ft mice in the 3rd week after exercise almost returned to the status of the 4th week of running exercise (Fig. 6e, f). However, the counterparts of wt/wt mice differed from each other at the 3 time points during and after exercise (Fig. 6e, f). Our results indicated that the gut microbiota of wild-type mice was more sensitive to exercise than that of $Asmt^{ft/ft}$ mice.

### $Asmt^{ft/ft}$ promotes SCFA production to suppress gut microbial adaptations to exercise.

Short-chain fatty acids (SCFAs) are important metabolites of gut microbes and act as signaling molecules to regulate the physiological activities of the host, so we next measured fecal concentration of SCFAs using GC–MS. A total of 7 SCFAs were detected. Two-way ANOVA showed that there was a significant increase in total SCFAs in the ft/ft mice, and a reduction with running exercise (Fig. 7a). Analysis of SCFA levels in the stool of the $Asmt^{ft/ft}$ mice revealed an increase in acetate ($p < 0.01$, Fig. 7b), propionate ($p < 0.01$, Fig. 7c), iso-butyrate ($p < 0.01$, Fig. 7d), isovalerate ($p < 0.05$, Fig. 7f), valerate ($p < 0.01$, Fig. 7g), and hexanoate ($p < 0.05$, Fig. 7h) compared to the wt/wt mice. Running exercise only reduced acetate ($p < 0.01$, Fig. 7b) and isobutyrate content ($p < 0.01$, Fig. 7d). Butyrate content was increased after exercise in the ft/ft group ($p < 0.01$, Fig. 7e). In all, $Asmt^{ft/ft}$ increased SCFA production and intra-group variation, thus leading to decreased gut microbial adaptations to exercise.

### Discussion

Women are more likely to suffer from depression than men, suggesting that women may have different susceptibility genes from men[23]. Genetic polymorphisms in circadian genes influence the risk of depression in a gender-dependent and stress-dependent manner[24]. Here, we identified *Asmt* as a potential gene for female susceptibility to anxiety and depression. Upon the *Asmt* frameshift mutation induced by CRISPR/Cas9 in *C57BL/6J* mice, female mice exhibited anxiety and depression-like behaviors, but no such abnormality was observed in male mice (Fig. 1a–g). Thus, the *Asmt* in wild-type *C57BL/6J* mice is one of

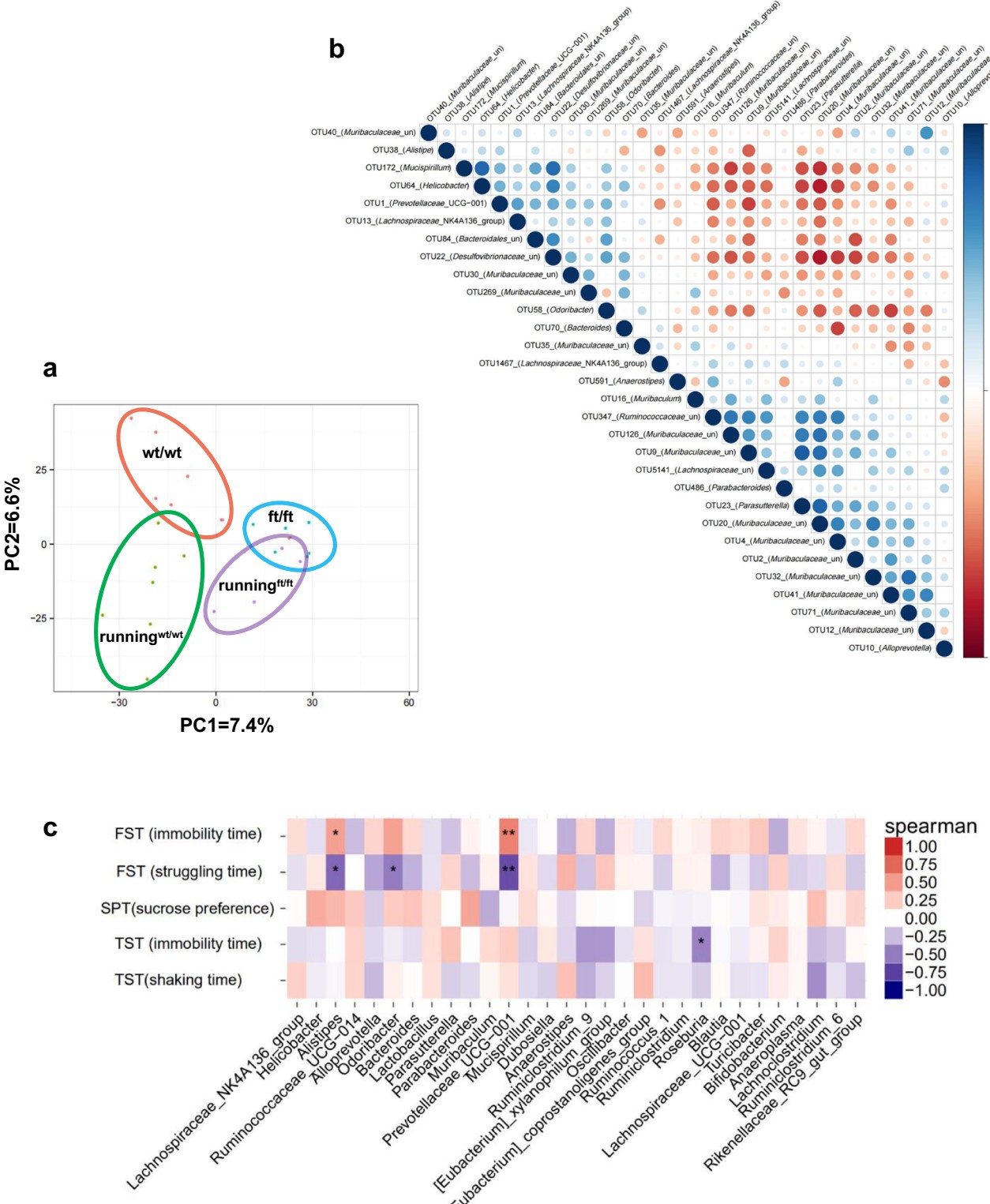

**Fig. 4 *Asmt*ft/ft reconstituted gut microbiota and resulted in lower adaptations to exercise. a** PCoA revealed a trend of separation by genotype (wt/wt, $n = 6$/group; ft/ft, $n = 5$/group), but running exercise induced a smaller trend of separation in *Asmt*ft/ft mice than in wild-type. **b** Relationship of OTUs with different abundances. In the top 30 species, Mucispirillum (OTU172), as well as Lachnospiraceae (OTU13), was negatively correlated with 14 OTUs and positively correlated with 10 OTUs. **c** Correlation of different species with behavioral indices. *$p < 0.05$, **$p < 0.01$.

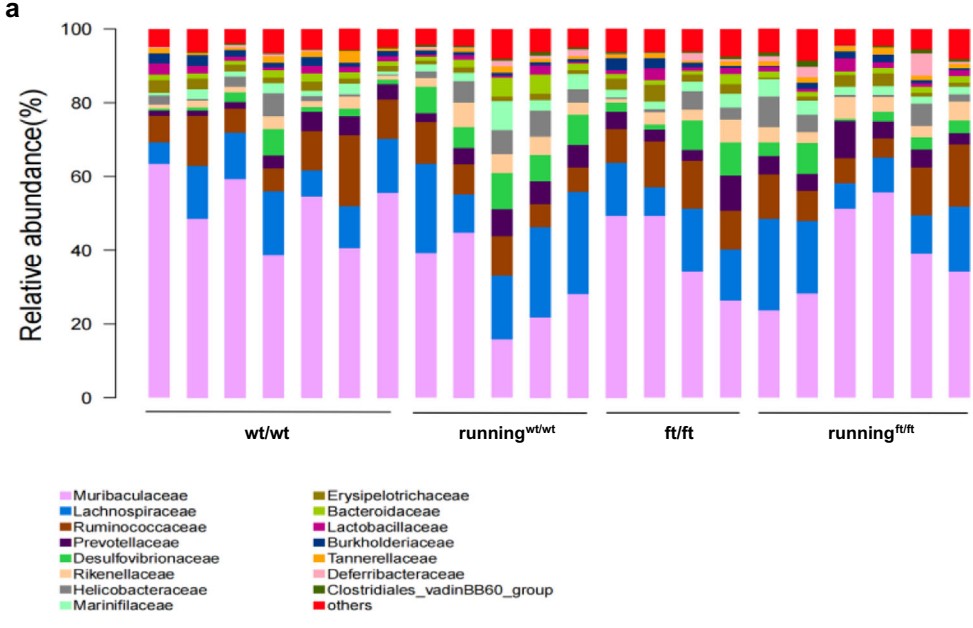

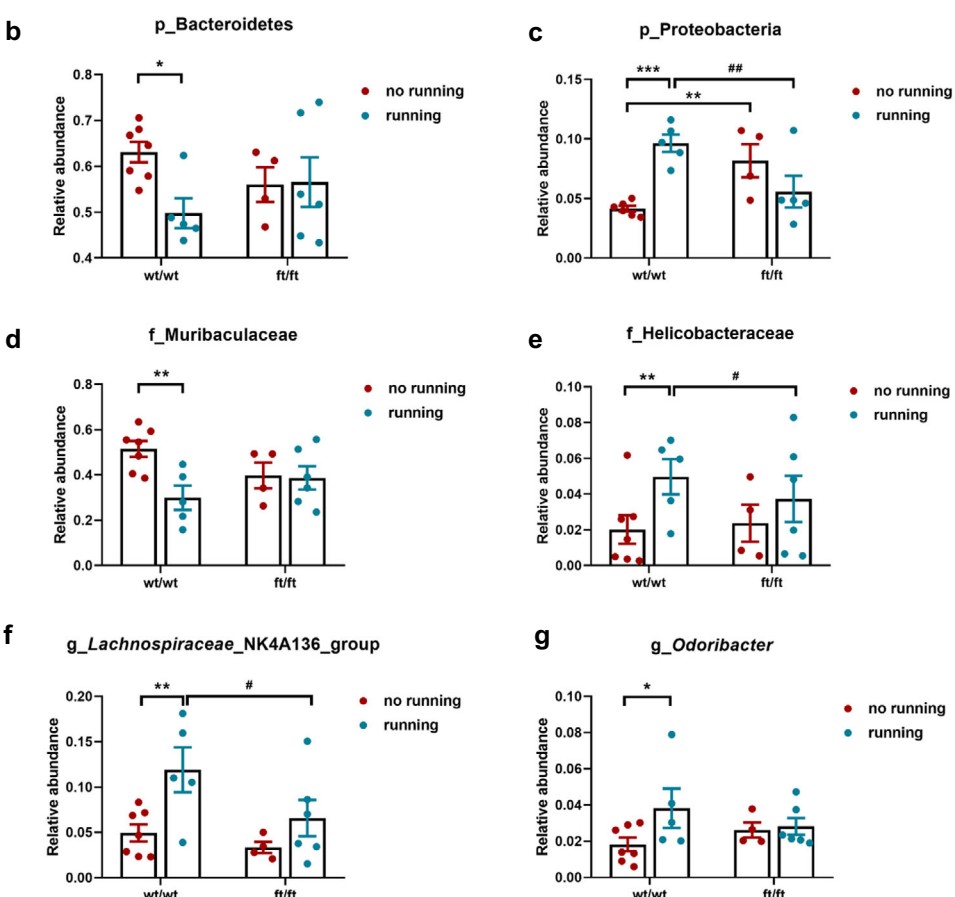

**Fig. 5 *Asmt*ᶠᵗ/ᶠᵗ attenuates the reconstitution of gut microbiota by exercise at multiple levels. a** Microbiotal distribution at the species level. In this case, the other diagrams at phylum, species, genus, and order level are omitted. **b**, **c** The relative abundance of Bacteroidetes and Proteobacteria at the phylum level ($n = 4$–7/group; **b** Two-way ANOVA, $F(1, 18) = 3.105$, $p = 0.0950$; **c** Two-way ANOVA, $F(1, 16) = 17.93$, $p = 0.0006$). **d**, **e** The relative abundance of Muribaculaceae and Helicobacteraceae at the family level ($n = 4$–7/group; **d** Two-way ANOVA, $F(1, 18) = 4.336$, $p = 0.0519$; **e** Two-way ANOVA, $F(1, 18) = 0.5531$, $p = 0.4667$). **f**, **g** The relative abundance of Lachnospiraceae and Odoribacter at the genus level ($n = 4$–7/group; **f** Two-way ANOVA, $F(1, 18) = 8.662$, $p = 0.0087$; **g** Two-way ANOVA, $F(1, 18) = 2.009$, $p = 0.1734$). Data are mean ± SEM. *$p < 0.05$, **$p < 0.01$; #$p < 0.05$, ##$p < 0.01$; Two-way ANOVA followed by post hoc test.

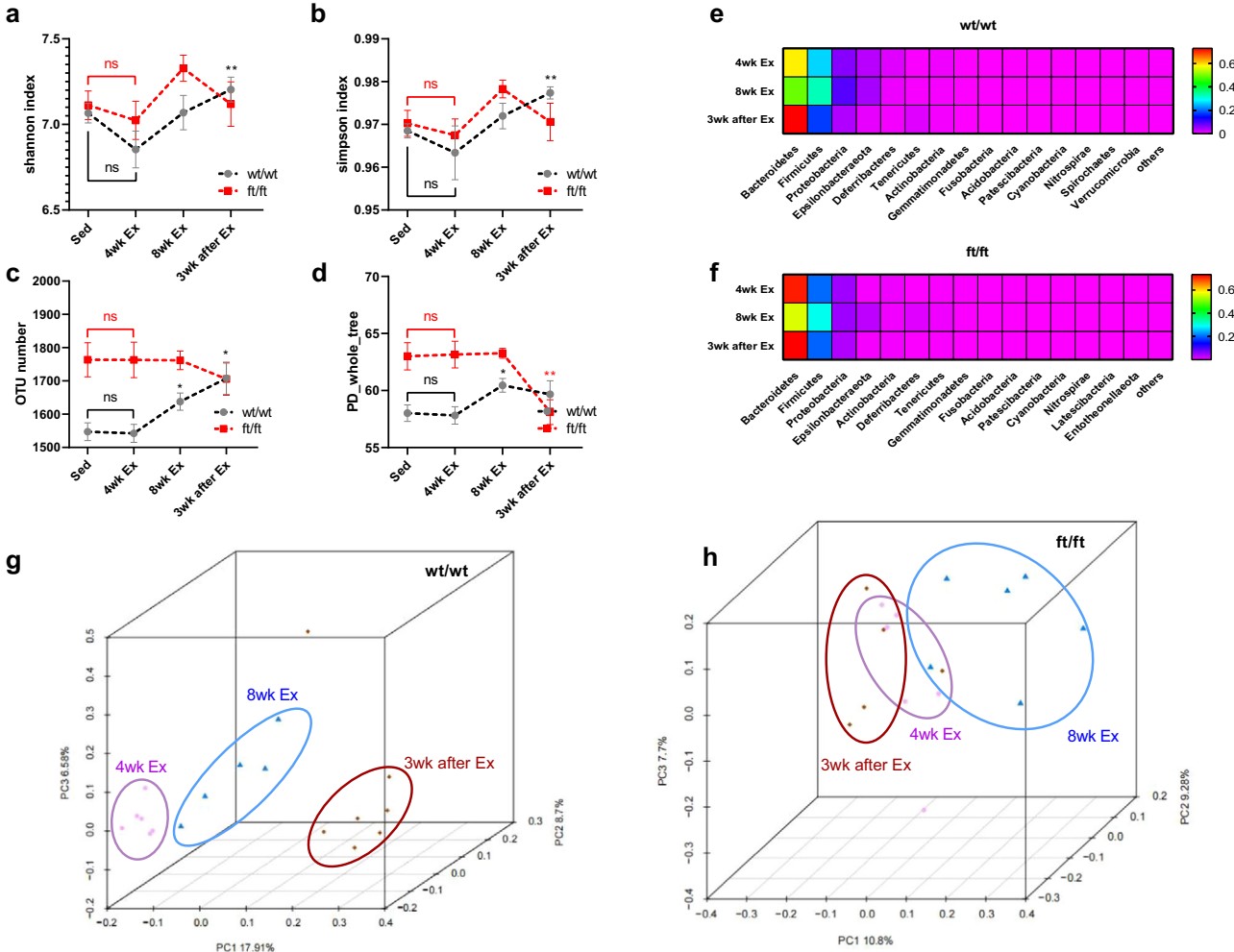

**Fig. 6 *Asmt*<sup>ft/ft</sup> suppressed gut microbial adaptations to exercise. a–d** The longitudinal changes of Shannon index, Simpson index, OUT number and PD whole tree before and after exercise (wt/wt, $n = 5–7$/time point; ft/ft, $n = 5–6$/time point; **a** Two-way ANOVA, $F(3,28) = 3.721$, $p = 0.0360$; **b** Two-way ANOVA, $F(3,28) = 3.914$, $p = 0.0335$; **c** Two-way ANOVA, $F(3,28) = 5.662$, $p = 0.0037$; **d** Two-way ANOVA, $F(3,28) = 6.932$, $p = 0.0012$). weeks effect: $*p < 0.05$, $**p < 0.01$. Two-way ANOVA followed by post hoc test. **e**, **f** Heatmap about the top 15 phyla showed the gut microbiota profile during and after exercise. **g**, **h** PCoA revealed an obvious trend of separation in gut microbiota in wild-type mice (**g**) but not in *Asmt*<sup>ft/ft</sup> mice (**h**). wt/wt, $n = 5–7$/time point; ft/ft, $n = 5–6$/time point.

the genes susceptible to depression and anxiety in females, even though it contains two mutated loci that lead to congenital deficiency in melatonin synthesis. The decrease in melatonin signaling may be one of the natural selections for laboratory mouse strains because melatonin inhibits reproduction and alters circadian rhythms[25]. Only in this way can mice meet the requirements of being a laboratory animal. Additionally, our behavioral results showed that *Asmt*<sup>ft/ft</sup> mice had lower adaptability to regular exercise than wild-type (Fig. 2a–g).

In the *C57BL/6J* mice, a gene mutation that was once thought to be insignificant caused anxiety- and depression-like behavior, and only in females. Why? We initially attributed this to melatonin deficiency, since ASMT is one of the key rate-limiting enzymes that have been identified for melatonin synthesis. Our results showed that *Asmt*<sup>ft/ft</sup> did not lead to a significant decrease in serum melatonin, as melatonin is naturally lower in *C57BL/6J* mice. *Asmt* that already has two mutated sites may not mind missing another 20 bp of exon 2. Therefore, neurobehavioral abnormalities in female *Asmt*<sup>ft/ft</sup> mice could not be attributed to melatonin deficiency. Gender is a priority because *Asmt*<sup>ft/ft</sup> causes anxiety- and depression-like behavior only in females (Fig. 1a–g), suggesting that *Asmt* remains important in females. Previous

studies have shown sex differences in the phenotype of knockout mice. Compared with wild-type and heterozygous mice, food-anticipatory activity was attenuated in dopamine D1 receptor knockout mice, with females showing greater attenuation than males[26]. Immobility time in TST was reduced in both male and female 5-HT receptor knockout mice after fluoxetine treatment, whereas females showed gender-related disinhibition of 5-HT release that sustained higher levels of hippocampal 5-HT and behavioral vulnerability to 5-HT depletion[27]. Not all these genes are on the sex chromosomes, but the knockout models show female susceptibility.

In this study, swimming as a customized exercise protocol for rodents reduced the distance and time in a central area in the OFT of wild-type mice (Fig. 2a, c). Moreover, the immobility time was increased and the struggle time was reduced in FST and TST (Fig. 2d–g). These results suggest that exercise may cause anxiety- and depression-like behaviors in *C57BL/6J* mice, which is contrary to our expectations and previous studies[28]. Although *Asmt*<sup>ft/ft</sup> mice showed anxiety- and depression-like behaviors, these behaviors did not increase or decrease under the stimulation of water. Judging from these behavioral changes, *Asmt*<sup>ft/ft</sup> mice were resistant to exercise, while the wild-type was more sensitive. It is

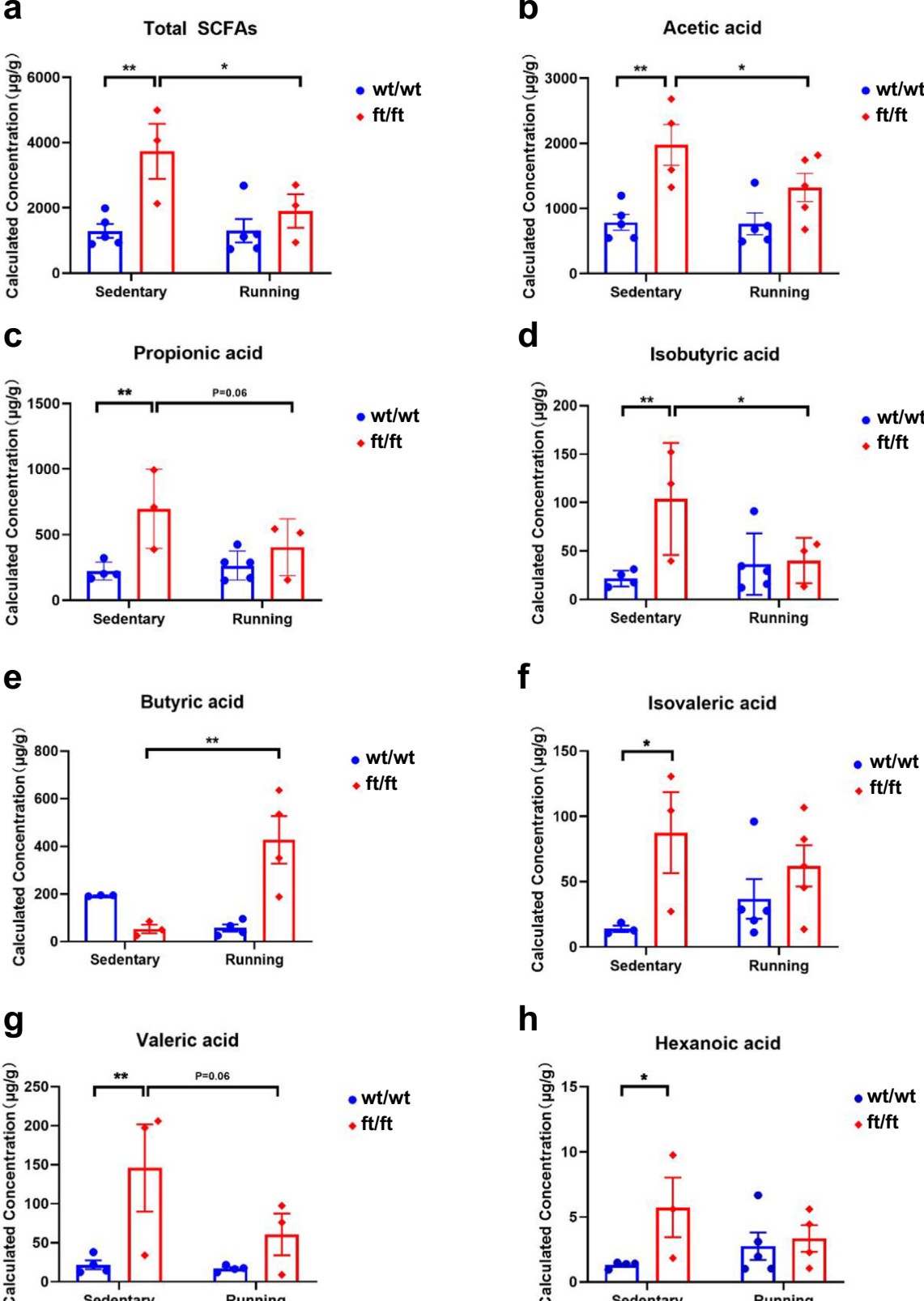

**Fig. 7 Asmt^{ft/ft} suppressed gut microbial adaptations to exercise by modulating SCFAs. a** Total SCFAs ($n = 3$–5/group; Two-way ANOVA, $F_{(1,12)} = 11.19$, $p < 0.01$). **b–h** Variations of each SCFA ($n = 3$–5/group; **b** Acetic acid, Two-way ANOVA, $F_{(1,15)} = 18.24$, $p < 0.01$; **c** Propionic acid, Two-way ANOVA, $F_{(1,11)} = 10.94$, $p < 0.01$; **d** Isobutyric acid, Two-way ANOVA, $F_{(1,11)} = 6.001$, $p = 0.0323$; **e** Butyric acid, Two-way ANOVA, $F_{(1,10)} = 3.729$, $p = 0.0823$; **f** Isovaleric acid, Two-way ANOVA, $F_{(1,12)} = 7.095$, $p < 0.05$; **g** Valeric acid, Two-way ANOVA, $F_{(1,10)} = 10.28$, $p < 0.01$; **h** Hexanoic acid, Two-way ANOVA, $F_{(1,12)} = 4.395$, $p = 0.0579$). Data are mean ± SEM. *$p < 0.05$, **$p < 0.01$; Two-way ANOVA followed by post hoc test.

widely accepted that the health benefits of exercise are genotype-dependent[29,30]. This is supported by our study. It is important to note that compulsive, even regular, swimming is chronic stress like FST for rodents, so swimming probably exacerbates depressive behavior. Our behavioral results show that $Asmt^{ft/ft}$ causes anxiety- and depression-like behaviors, and also resists the chronic stress of "swimming" as a forced exercise. The reason was that mice were not sensitive to external stimuli after genotype change.

We determined here that $Asmt^{ft/ft}$ mice had a completely different gut microbiome structure compared to wild-type. A recent study showed that ingestion of *Faecalibaculum rodentium* led to depression-like phenotypes in mice through the brain–gut–microbiota axis[31]. Similar to our study, female *cynomolgus macaque* exhibited naturally depression-like behavior, and gut microbial structure and metabolic characteristics of depression-like macaques were different from those of the control group (the gut microbiota of depressed monkeys also had a disturbance of *Firmicutes*)[32]. In this study, we identified a series of dominant species enriched in $Asmt^{ft/ft}$ mice, including *Faecalibaculum rodentium, Firmicutes, Lactobacillus salivarius*. In $Asmt^{ft/ft}$ mice, we observed a reduced abundance of *Mucispirillum schaedleri* and *Desulfovibrionales* (Figs. 5a and S5), which have been shown to improve inflammation in stress and ACTH-induced depression[33,34]. These findings may explain why we observed anxiety- and depression-like behaviors in $Asmt^{ft/ft}$ mice.

In addition, it is necessary to explain why the behaviors of $Asmt^{ft/ft}$ mice exhibit lower adaptability to exercise. A cross-sectional comparison of gut microbiota suggests a possibility. First, ft/ft and wt/wt mice had completely separate gut microbiota structures at the baseline level, as was evidenced by the PCoA analysis in Fig. 4a. Second, we found that exercise led to a separation of gut microbiota structure in wild-type (Fig. 4a), that is to say, exercise-induced a new gut microbiota structure in wt/wt mice. In contrast, exercise induced a much smaller deviation in the gut microbiota of $Asmt^{ft/ft}$ mice (Fig. 4a). Most studies have suggested that exercise benefits may be determined by the structural remodeling of the gut microbiota. Fermentation of gut microbiota determines the effect of exercise on diabetes prevention. The patients whose microbiota is sensitive to exercise showed an increased capacity for SCFAs biosynthesis and branched-chain amino acid catabolism, whereas insensitive patients were characterized by the production of metabolically harmful compounds[35]. SCFAs metabolism not only affects the exercise benefit to diabetes but also has a close relationship with depression and the brain. A recent study showed that *Bacillus licheniformis* increased SCFA levels in the colon and altered levels of neurotransmitters in the brain, thereby preventing depression and anxiety-like behaviors[36]. The microbiota–gut–brain axis is responsible for a variety of metabolic, neurodegenerative, and neuropsychiatric diseases. The metabolites of the gut microbiome can induce epigenetic modifications through DNA methylation, histone modification, and non-coding RNA-associated gene silencing. SCFAs, especially butyrate, are well-known inhibitors of histone deacetylase[37]. Our further analysis of fecal SCFAs showed an interaction similar to that of the gut microbiota, in which recombination of SCFAs associated with genotype resulted in an insensitivity of SCFAs to exercise (Fig. 7a). In all, we suggested that $Asmt^{ft/ft}$ impaired the sensitivity of mice gut microbiota to exercise.

In addition, longitudinal analysis showed that the gut microbiota structure of wild-type mice shifted greatly during exercise, which was characterized by the highest abundance of *Lactobacillus murinus* in the 4th week, *Lactobacillus gasseri* in the 8th week and *Bacteroidales bacterium* in the 3rd week after the exercise (Fig. 6e). This may determine neurobehavioral

adaptability to exercise in mice. In contrast, the gut microbiota of $Asmt^{ft/ft}$ changed little during exercise, and the structure of the gut microbiota at the phylum level in the 4th week of exercise was highly similar to that in the 3rd-week post-exercise (Fig. 6f). This further explains why the anxiety- and depression-like behaviors of $Asmt^{ft/ft}$ mice were not improved by exercise. The longitudinal changes in the gut microbiota during exercise further warn us that fecal collection should be at the same timepoint (Fig. 6g and h). Inter-group comparisons of the gut microbiota of feces collected during different exercise weeks were not reliable.

There are some notable limitations to this study. First, spontaneous behaviors vary widely between individual animals, and any significant findings related to genotype or sex–genotype interactions are likely to be false positives. Although mixed-sex comparisons and chi-square tests highlighted the greater effects of *Asmt* mutations on female mice, such limitations are now hard to avoid. This study sheds new light on gender differences in depression. That is, *Asmt*, an X chromosome gene that seems useless to *C57BL/6J* mice, has a potential effect on the behavior of female mice. Second, the sample size of some behavioral data is too small, because not enough mice were genetically identified for testing or behavioral tests had missing or abnormal data records. Finally, we must admit that this study did not notice natural mutations of *Asmt* in *C57BL/6J* mice at the gene targeting phase. This mutation caused lower melatonin levels in the *C57BL/6J* mice, and further frameshift mutations did not reduce melatonin levels, so the melatonin hypothesis for depression should not be convincing here.

In conclusion, our data indicate that naturally mutated *Asmt* is not useless but still critical for neurobehavioral adaptability in *C57BL/6J* mice during exercise. Further knockout of *Asmt* did not affect serum melatonin levels, but led to anxiety- and depression-like behaviors and a structural modulation of gut microbiota in females. Exercise did not improve anxiety and depression in $Asmt^{ft/ft}$ genotype, which may be the outcome of exercise not changing the gut microbiota. *Asmt* is associated with neurobehavioral adaptations to exercise in females. These findings could help develop strategies targeting anxiety and depression in women.

## Methods

**Animals**. $Asmt^{ft/ft}$ mice were generated by Shanghai Humangen Biotech Inc. using CRISPR-Cas9. The intron–exon organization of the 5411 bp *Asmt* gene was manually annotated from the current build of the *C57BL/6J* mouse genome (Supplementary Table S1). Five sgRNAs targeted to DNA sequences within exons 2 and 3 were designed (Fig. S6a, Supplementary Table S2), with the aim of generating frameshift mutation of *Asmt* ($Asmt^{ft/ft}$). Gene sequencing showed that 2 sgRNAs have higher deletion efficacy (Fig. S6b). sgRNA 1 and 2 led to exon 2 deletion, as shown by the presence of smaller molecular weight bands (Fig. S6c, lanes 5 and 6). The two sgRNAs were used for micro-injection of embryonic stem cells in *C57BL/6J* mice. The founder mouse lacking 20 bp had an obvious frameshift mutation at the translational level, so it was used to cross with a wild-type female and generate $Asmt^{ft/ft}$ mouse strain (Fig. S6d). All procedures were in accordance with Laboratory Animal Guidelines for Ethical Review of Welfare published by the General Administration of Quality Supervision, Inspection and Quarantine of the People's Republic of China (GB/T 35892-2018, February 2018), and were approved by the Experimental Animal Care and Use Committee at East China Normal University (m2019227, February 2019). We have complied with all relevant ethical regulations for animal testing.

**Experimental design**. Mice were genotyped at 3 weeks of age and grouped according to sex and genotype at 4 weeks of age. The wt/wt and ft/ft mice were numbered and kept in separate cages and were subjected to regular swimming or running at 6 weeks of age. Feces were collected for gut microbiota analysis during the 4th and 8th week of exercise and in the third week after exercise (i.e., after behavioral tests). Behavioral tests were performed at 15–17 weeks of age, and sacrifices were made at 18 weeks of age (Fig. 1i). Daily swimming as a common regular exercise for mice was performed in a glass water tank ($100 \times 60 \times 80$ cm) at $32 \pm 1 \,°C$. Formal training began at the age of 6 weeks, with intensity of 60 min/day, 6 days/week, for 8 weeks. It is worth emphasizing that this is quite different from the forced swim test below. FST was performed in a smaller space with cold water that was very unfriendly to animals, and the purpose was to observe the struggling and resistance behaviors of mice. Treadmill exercise was conducted in a separate study. Formal training consisted of 40 min/day running at 13 m/min, 5 days/week, for 8 weeks.

**Behavioral tests**. All testing equipment was thoroughly cleaned between each session. Except sucrose preference test (SPT), all groups of mice underwent parallel behavioral tests in the daytime 9:00–12:00 a.m. For acclimation, mice were housed in the testing room (a black box with sound insulation) 30 min prior to the start of each behavioral test. The black box was lit with indirect illumination by 250 lx for behavioral tests. A computerized video tracking system (DigBehav, Jiliang Go. Ltd., Shanghai, China) was used to record mouse behaviors in OFT, FST, and TST. The sucrose preference test took a week, so OFT, FST, and TST were scheduled for the following week (Fig. 1i).

The sucrose preference test (SPT) was performed as we described previously[28,38]. Briefly, 72 h before the test mice were trained to adapt 1% sucrose solution (w/v): two bottles of 1% sucrose solution were placed in each cage, and 24 h later 1% sucrose in one bottle was replaced with tap water for 24 h. After adaptation, mice were deprived of water and food for 24 h, followed by the sucrose preference test, in which mice housed in individual cages had free access to two bottles containing 200 ml of sucrose solution (1% w/v) and 200 ml of water, respectively. At the end of 24 h, the sucrose preference was calculated as a percentage of the consumed 1% sucrose solution relative to the total number of liquid intake.

The open-field test (OFT) was used to examine anxiety-like behaviors in mice as we described previously[28]. Place each mouse in the open field center ($30 \times 30 \times 30$ cm chamber, with 16 holes in its floor) for 5 min in a quiet room. Mice that prefer staying close to the walls and travel more in the periphery can be described as showing thigmotaxis (movement towards a solid object), pronounced in mice showing signs of anxiety-like behavior. Mice with lower anxiety tend to spend more time in the central, open area of the box[39].

The forced swim test (FST) remains one of the most used tools for screening antidepressants since it has good predictive validity and allows rapid and economical detection of substances with potential antidepressant-like activity[40]. The swimming sessions were conducted by placing the mice in cylinders (30 cm height × 10 cm diameter) containing $25\,°C$ water 20 cm deep so that the mice could not support themselves by touching the bottom with their feet. The FST was conducted for 5 min and immobility time was recorded. Floating in the water without struggling and only making movements necessary to keep its head above the water were regarded as immobility. The FST was well established to assess a classical depressive behavior—despair behavior in mice as we described previously[28,38].

The tail suspension test (TST) was performed as we described previously[28,38]. Mice were individually suspended by the tail to a vertical bar on the top of a box ($30 \times 30 \times 30$ cm), with adhesive medical tape affixed 2 cm from the tip of the tail. The immobility time was recorded for a 5 min session. In the TST, immobility was defined as the absence of any limb or body movements except those caused by respiration.

**Immunofluorescence**. Fresh tissue was washed and fixed with 4% paraformaldehyde for 12 h or more, and then transferred into a 15% sucrose solution for 8 h and subsequently into a 30% sucrose solution overnight for dehydration. The frozen sections were dried at room temperature, fixed in 4% paraformaldehyde (DEPC) for 10 min, and shaken on the shaker in PBS (pH 7.4) three times for 5 min each. The frozen sections were incubated with a hybridization buffer at $37\,°C$ for 1 h. After removing the hybridization solution, sections were washed for 10 min at 37. Formamide washing can be added if there are more non-specific hybrids. Blocking serum was added to the section and incubated at room temperature (RT) for 30 min. PBS containing 1:200 dilution of primary antibody (GAP43, GB11095, Servicebio; NeuN, GB11138, Servicebio; GFAP, GB11096, Servicebio; Synaptophysin, GB11553, Servicebio) were incubated with sections at $4\,°C$ overnight. Sections were then washed with PBS three times for 5 min each at RT. After washing, the section was incubated for 50 min with a second antibody at RT. Samples were then washed with PBS three times for 5 min each at RT. Incubated with DAPI for 8 min in the dark, the section was then mounted with anti-fluorescence quenching sealing tablets. By the microscopic examination and photography, DAPI glows blue by UV excitation wavelength 330–380 nm and emission wavelength 420 nm; Cy3 glows red by excitation wavelength 510–560 nm and emission wavelength 590 nm. Dorsal hippocampus (CA1d), ventral hippocampus (CA1v), and hippocampal dentate gyrus (DG) are functionally distinct structures. Spatial memory mainly depends on CA1d, whereas CA1v relates to stress and emotion[41–43]. The activity and plasticity of DG are closely related to chronic stress, learning and memory[44,45]. Therefore, these regions were analyzed separately.

**Transmission electron microscopy (TEM)**. Fresh tissues should be selected to minimize mechanical damage such as pulling, contusion and extrusion. Tissue blocks could be removed from the animal body and immediately put into petri dishes, and then cut into small sizes of 1 mm³ in the fixative. The 1 mm³ tissue blocks were transferred into an EP tube with fresh TEM fixative for further fixation, which was fixed at $4\,°C$ for preservation and transportation. Tissues were next fixed with 1% $OsO_4$ in 0.1 M PB (pH 7.4) for 2 h at room temperature. After removing $OsO_4$, the tissues are rinsed in 0.1 M PB (pH 7.4) for 3 times and then dehydrated at RT using 30%–100% ethanol. With resin penetration and embedding, samples were moved into $65\,°C$ oven to polymerize for more than 48 h. The resin blocks were cut to 60–80 nm thin on the ultramicrotome, and the tissues were fished out onto the 150 meshes cuprum grids with formvar film. Slices were stained with 2% uranium acetate for 8 min, rinsed in 70% ethanol for 3 times, rinsed in ultra-pure water for 3 times, stained with 2.6% Lead citrate for 8 min, and then rinsed with ultra-pure water for 3 times. Dried by the filer paper, the cuprum grids were put into the grids board and dried overnight at RT. The cuprum grids were observed under TEM and taken images.

**Fecal DNA extraction and *16S* sequencing**. All fecal samples were freshly collected and stored at $-80\,°C$ until analysis. Fecal DNA was extracted using E.Z.N.A. Stool DNA Kit (Omega Bio-tek, Inc., GA). The primers F1 and R2 (5'-TACGGRAGGCAGCAG-3' and 5'-AGGGTATCTAATCCT-3') correspond to positions

343–798 in the *Escherichia coli 16 S* rRNA gene were used to amplify the V3–V4 region of each sample via PCR. PCR products from all samples were sequenced on the Illumina Miseq platform (Illumina, Inc., USA) using a Miseq Reagent Kit V3 (600-cycle, MS-102-3033, Illumina, USA) based on the manufacturer's instructions (Illumina, USA). *16S* rRNA gene encodes the small subunit rRNA of the prokaryotic ribosome, which is about 1542 bp in length and contains 10 conserved regions and 9 hypervariable regions. *16S* rRNA sequencing was used as a method for bacterial classification in this study because *16S* rRNA has the following advantages: it is widely present in all prokaryotes, has sufficient phylogenetic information, is easy to amplify, and has a comprehensive information database[46].

**Bioinformatics and statistical analysis of *16S* sequencing data**. Bioinformatics analysis was provided by SHANGHAI BIOTREE BIOTECH with reference to previous studies[47]. There were 71 samples in this study. After quality control (QC value = 20), clean tags are 38,021–41,962. After removing the chimera, valid tags (the data finally used for analysis) are 34608–39225. The average length of valid tags is 412.09–419.92 bp, and the number of OTU in each sample is 1475–1941. The non-parametric Mann–Whitney $U$-test was used to test for differences between the two groups. Both weighted and unweighted UniFrac values were calculated in QIIME. The QIIME pipeline was also used to generate principal coordinate analysis (PCoA) plots for visualization of the unweighted UniFrac dissimilarity. Species annotation analysis was performed on the Silva database (http://www.arb-silva.de/) by the Mothur algorithm to study the difference of dominant species and community composition in different samples. The linear discriminant analysis (LDA) effect size (LEfse) was used to detect taxa with differential abundance among groups. LEfSe analysis was performed under the following conditions: (1) the alpha value for the factorial Kruskal–Wallis test among classes is <0.05 and (2) the threshold on the logarithmic LDA score for discriminative features is >2.0.

**GC–MS analysis for SCFAs**. 50 mg fecal sample was mixed with 0.5 mL $dH_2O$ by vortex for 10 s, and homogenized for 4 min grinding at 40 Hz and for 5 min ultrasound treatment in ice water. Homogenate was centrifuged for 20 min at 5000 rpm, 4 °C. 0.3 mL supernatant was transfered into EP tubes, vortex mixing with 0.5 mL $dH_2O$ for 10 s. Mixture was homogenized for 4 min at 40 Hz and for 5 min ultrasound treatment in ice water again. After centrifuging for 20 min at 5000 rpm, 4 °C. 0.5 mL supernatant was transferred into a fresh EP tube, combined with the above 0.3 mL supernatant. 0.1 mL 50% $H_2SO_4$ and 0.8 mL of 2-Methylvaleric acid (25 mg/L stock in methyl tert-butyl ether) were added into a total of 0.8 mL supernatant as internal standard, vortex mixing for 10 s, oscillations in 10 min, then ultrasound treatment for 5 min in ice water. Centrifuge for 15 min at 10,000 rpm, 4 °C and keep at −20 °C for 30 min, the supernatant was transferred into a glass vial for GC–MS analysis (SHIMADZU GC2030-QP2020 NX).

1 μL aliquot of the analyte was injected in split mode (5:1). Helium was used as the carrier gas, the front inlet purge flow was 3 mL min$^{-1}$, and the gas flow rate through the column was 1 mL min$^{-1}$. The initial temperature was kept at 80 °C for 1 min, then raised to 200 °C at a rate of 10 °C min$^{-1}$ for 5 min, then kept for 1 min at 240 °C at a rate of 40 °C min$^{-1}$. The injection, transfer line, quad and ion source temperatures were 240, 240, 200 and 150 °C. The energy was −70 eV in electron impact mode. The data were acquired in Scan/SIM mode with the *m/z* range of 33–150 after a solvent delay of 3.5 min.

**Statistics and reproducibility**. All statistical analyses were performed using Graphpad prism 8.01. The data were expressed as mean ± SEM. Two-way ANOVA and post-hoc were used for assessing differences between groups including sex × genotype and exercise × genotype, and one-way ANOVA was used for longitudinal comparisons of gut microbiota. Nested $t$-test and Chi-square test were used for mixed-sex comparisons on the behavioral data. Differences were considered statistically significant when $p < 0.05$.

**Reporting summary**. Further information on research design is available in the Nature Portfolio Reporting Summary linked to this article.

## Data availability

The data that support the findings of this study are openly available in the NCBI database at https://submit.ncbi.nlm.nih.gov/subs/bioproject/SUB13860403/overview, BioProject ID: PRJNA1022313. The QIIME2 tables underlying the analyses are available in Supplementary Data 1. Source data related to plots and graphs in the manuscript can be found in Supplementary Data 2. All other data are available from the corresponding author upon reasonable request.

## Code availability

The analysis methods and software used in this article are open-source and do not generate new codes.

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

## Acknowledgements

This work was supported by the National Natural Science Foundation of China (32271174 and 31871208) and the Shanghai Natural Science Foundation (18ZR1412000). We thank Dr. Xiujie Yuan, the director engineer of Shanghai Humangen Biotech Inc., for designing SgRNA-guided *Asmt* frameshift mutation and generating the founder mouse for this study. We thank the director engineers in Shanghai OE Biotech Co., Ltd and Biotree Biomedical Technology Co., Ltd, for their expertise and conversations about *16S* rRNA sequencing.

## Author contributions

W.L(Weina Liu). and Z.Q. conceived, designed, performed, and interpreted experiments and made figures. Z.H., Y.Z. (Ye Zhang), S.Z., W.L. (Wenbin Liu), L.L., Z.C., and Y.Z. (Yong Zou) performed animal care, exercise, genotyping, behavioral tests, real-time PCR, Western blot, and data collection, and made figures. J.X. participated in interpreting results and supervising the experimental plan. W.L. (Weina Liu) and Z.Q. wrote and revised the manuscript.

## Competing interests

The authors declare no competing interests.
