## [Peer Review File · Communications Biology]

Reviewers' comments:

Reviewer #1 (Remarks to the Author):

Liu and colleagues have generated a mouse in which the *Asmt* gene is rendered non-functional through a CRISPR-induced genetic mutation causing a frameshift. They have compared male and female genetically-altered mice to their wildtype counterparts on a variety of behavioural assays, with respect to measures of synaptic plasticity using immunofluorescence/microscopy and with respect to the composition of gut microbiota.

There is certainly a need for a comprehensive phenotypic characterisation of the ASMT-deficient mouse in order to shed light upon the potential functions of ASMT and the consequences of its loss. However, this study has several critical flaws which severely limit its utility.

The authors suggest that their approach may shed light upon depression-related phenotypes in humans, although there is little prior evidence supporting an association between ASMT deficiency and mood disorders in humans. The experimental approaches the authors use are not adequately justified - it is not clear why the specific behavioural tests have been selected, why synaptic plasticity may be a particularly informative neurobiological measure to assess, nor why gut microbiota should be affected by the genetic mutation. There is little detail provided about the conditions under which the behavioural experiments were conducted (apparatus, light levels, cleaning protocols etc.). The analyses of spontaneous behaviours, where outcome measures can vary considerably between individual animals, are severely underpowered (especially as groups are stratified by sex), and, as such any significant findings relating to genotype or sex x genotype interactions are highly likely to be false positives.

Reviewer #2 (Remarks to the Author):

In this study, the authors deleted 20bp of exon 2 of *Asmt* by CRISPR/Cas9, and accidentally generated C57BL/6J mouse with ASMT frameshift mutation (ASMTft/ft). The wt/wt and ft/ft mice were subjected to regular swimming or running at 6 weeks of age. Feces were collected for gut microbiota analysis at the 4th and 8th week of exercise, and behavioral tests were performed at 15-17 weeks of age. Immunofluorescence, transmission electron microscopy, and 16S sequencing were employed. The results showed that female ASMTft/ft mice exhibited anxiety- and depression-like behaviors compared to wildtype, accompanied by structural modulation of gut microbiota. ASMTft/ft mice showed a lower neurobehavioral response to exercise, while wildtype showed a "higher response". Cross-sectional and longitudinal comparisons showed that exercise resulted in less gut microbiota remodeling in ASMTft/ft mice than that in wildtype. The data of this study are interesting. However, the following concerns should be addressed.

- 1) Methods: Three behavioral tests such as OFT, FST, and TST were used. However, the Figure 4D has SPT data. Did you perform SPT for this study? If you have SPT data in the Figures 1 and 2, SPT data should be included as figures.
- 2) Data such as two factors should be analyzed using two-way ANOVA, and post-hoc ??? test. The results of statistical analysis should be included in the legends.
- 3) Sample size of some figures is too small.
- 4) Poor writing.

Reviewer #3 (Remarks to the Author):

An interesting paper addressing the overlooked theme of increased predisposition to female depression in an animal model.

A few housekeeping notes are provided below:

Line 62 - if the gene coordinates are known then they should be completed;

Line 81 - because the methodology is at the end of the manuscript, so abbreviations should be expanded when they first appear (OFT, FST, TST);

Line from 77 (results) - long and difficult to read results, it is worth supplementing with a simple table indicating with "arrows" the increase or decrease in e.g. physical activity or expression of the studied genes, microbiota diversity index in the compared subgroups male vs female and wt/wt vs ft/ft;

Line from 298 (method):

- complete publications or references to sources for the individual protocols described
 - complete the number and date of Commission approval for the study
 - justify the choice of microbiome analysis method (16S vs whole genome sequencing)
 - complete the minimum sequencing QC values allowing for further analysis
 - complete the reference database data used to annotate sequencing results
- Improve quality of figures - especially F4B - illegible taxa; F5B - truncated figures.

Reviewer #1 (Remarks to the Author):

Although the authors have included some additional information, I don't feel that this addresses my previous concerns - the rationale for the approaches is still very general, the approaches are poorly-justified, and simply stating that the findings are likely to be false positives as a limitation in the Discussion section is insufficient. The authors might consider rewriting the manuscript pooling sexes and comparing mixed sex wildtype vs. knockout animals.

Reviewer #2 (Remarks to the Author):

All comments have been addressed.

Reviewer #3 (Remarks to the Author):

The authors used the suggestions and supplemented important information. I have no additional comments.

Point by point response to Reviewer

Reviewer #1 (Remarks to the Author):

Although the authors have included some additional information, I don't feel that this addresses my previous concerns - the rationale for the approaches is still very general, the approaches are poorly-justified, and simply stating that the findings are likely to be false positives as a limitation in the Discussion section is insufficient. The authors might consider rewriting the manuscript pooling sexes and comparing mixed sex wildtype vs. knockout animals.

Response: Thanks for your suggestions. In response to your doubts about the methods, the explanation is as follows:

1. Mixed sex comparison erased behavioral differences in female mice.

All behavioral data were analyzed by nested t-test. Results showed no behavioral differences between wild-type and knockout animals, with P-values between 0.3489 and 0.9638, suggesting that these differences were not statistically significant. However, Chi-square test results showed that there were gender differences between OFT and FST indicators. The reason is that there are sex differences in some behavioral parameters in wild-type. When the sexes were mixed, these differences were erased. Therefore, we believe that this gene manipulation is more sensitive to female mice, and female mice should be studied in subsequent experiments.

The statistical results are as follows:

2. The genetic manipulation of frameshift mutations is unpredictable

Frameshift mutation manipulation of ASMT has not been studied before, so this study has no sex bias. The choice of female mice was made after behavioral tests. In addition, this study is the first to use CRISPR-Cas9 to target exons of ASMT genes, and the genetic manipulation of frameshift mutations is uncertain.

Briefly, mouse *Asmt* gene (ID: 107626) is located in the F5 region of X chromosome. The MOUSE *Asmt* gene has 8 exons, and all exons are involved in protein coding. Gene targeting on the 2nd and 3rd exon will cause frameshift mutation and eventually lead to gene inactivation, so a total of 5 SgRNAs were designed for exons 2 and 3, and the splicing efficiency was tested in NIH3T3 cells. The most efficient 1~2 SgRNAs were selected for prokaryotic injection with Cas9 mRNA to obtain Founder mice with *Asmt* frameshift mutation in the context of C57BL/6 mouse strain. After microinjection, the genetically targeted embryos were transferred into the uteruses of two surrogate mice. Three weeks later, two surrogate mothers gave birth to four Founder mice. Genotyping showed that mouse 2# was missing 20bp of exon 2, mouse 3# was missing 9bp of exon 2, and the other two were off-target. Finally, 2# mouse was crossed with wild-type C57BL/6 mouse to generate ASMT^{ft/ft} mouse series. F5 and F6 generation were genotyped and included in the current study.

The following are data from the genetic manipulation process, but are not appropriate to present as results in the main text.

Figure S1. *Asmt* gene structure and scheme of SgRNA-guided frameshift mutation.

(A) The *Asmt* gene contains 8 exons. In order to produce effective frameshift mutations in this gene, we designed five SgRNAs targeting exons 2 and 3 in the upstream of the gene (see Supplementary Table S2).

(B) *Asmt*-SgRNAs plasmid was transiently transfected into NIH3T3 cells, and the sequencing results showed that SgRNA1 and SgRNA2 had higher efficiency.

(C) Synthesis of Cas9 mRNA and Asmt-SgRNA1/2 transcripts in vitro. Lane 1, 6kD RNA marker; Lane 2, Cas9 mRNA (before adding a poly(A) tail); Lane 3, Cas9 mRNA (after adding a poly(A) tail); Lane 4, 1kD RNA marker; Lane 5, Asmt-SgRNA1; Lane 6, Asmt2-SgRNA2. Electrophoresis showed that Asmt-SgRNA1 and Asmt-SgRNA2 had high transfection efficiency in vitro.

(D) Gene sequencing of the 2# founder mouse showed that Asmt had a 20bp loss of exon 2, resulting in a highly efficient frameshift mutation.

Supplementary Table S1

Intron-exon organization of the murine Asmt, chrX:170,672,644-170,678,054 (base numbering in base pairs from mm10, Sept 2019 assembly of the mouse genome at

https://genome.ucsc.edu/cgi-bin/hgGene?hgg_gene=ENSMUST00000178693.1&hgg_prot=uc029xop.1&hgg_chrom=chrX&hgg_start=170672643&hgg_end=170678054&hgg_type=knownGene&db=mm10&hgssid=1134780985_SMbHyvni75QmDiKhDdA8YPv5DSTz).

Feature	Start	End	Size	
Exon 1	170,672,644	170,672,753	110	
Intron 1	170,672,754	170,673,624	871	
Exon2	170,673,625	170,673,796	172	sgRNA target
Intron 2	170,673,797	170,674,621	825	
Exon3	170,674,622	170,674,754	133	sgRNA target
Intron 3	170,674,755	170,674,960	206	
Exon4	170,674,961	170,675,029	69	
Intron 4	170,675,030	170,675,747	718	
Exon5	170,675,748	170,675,866	119	
Intron 5	170,675,867	170,676,337	471	
Exon6	170,676,338	170,676,493	156	
Intron 6	170,676,494	170,677,016	523	
Exon7	170,677,017	170,677,139	123	
Intron 7	170,677,140	170,677,745	606	
Exon8	170,677,746	170,678,054	309	

Supplementary Table S2

SgRNAs designed for gene targeting

SgRNA#	Sequence
Asmt -SgRNA1	5'-GGGCCGCGGCGTCTGAACACG-3'
Asmt -SgRNA2	5'-GCGGCGCTGGCGAGGTCGTC -3'
Asmt -SgRNA3	5'-GCCGCGTCCCCGGGGGCTC -3'

Asmt -SgRNA4* 5'-gCGCCTACACCAACTCCCCC -3'

Asmt -SgRNA5 5'-GCGCAGCCTCCTGCTTACC-3'

* **g** was added at the 5'-terminal for SgRNA4 to facilitate the activation of the hU6 promoter.

3. Further mutations in Asmt are not without physiological significance

We must admit that this study did not notice natural mutations of ASMT in C57BL/6J mice at the start. This mutation caused lower melatonin levels in the C57BL/6J mice, and further frameshift mutations did not reduce melatonin levels, so the melatonin hypothesis for depression should not be convincing here.

However, after the establishment of ASMT^{ft/ft} mice, we still found that the gene manipulation had a significant effect on the behavior of female mice. Although the melatonin hypothesis cannot be explained here, the structural recombination of gut microbiota gives us new insights. Can these insignificant genes really be knocked out? It seems that there are more unknown functions of the Asmt gene in C57BL/6J mice worth exploring.

In addition, gene targeting deletes only 20bp of exon 2, and it is difficult to predict whether Asmt frameshift mutation can lead to expression of a novel protein. The results presented so far in this study are all based on statistical analysis and present a new insight for peer discussion.

Finally, in response to the above problems, we have rewritten the main text. Please refer to the content marked in the revised manuscript.

Reviewer #2 (Remarks to the Author):

All comments have been addressed.

Reviewer #3 (Remarks to the Author):

The authors used the suggestions and supplemented important information. I have no additional comments.

REVIEWERS' COMMENTS:

Reviewer #1 (Remarks to the Author):

The authors have addressed my concerns to some extent and have conceded that some of their findings are likely to be false positives as a consequence of low power. The manuscript is significantly improved compared to the original submission.